# Interplay of OpdP Porin and Chromosomal Carbapenemases in the Determination of Carbapenem Resistance/Susceptibility in *Pseudomonas aeruginosa*

Jad Atrissi,[a] Annalisa Milan,[a] Raffaela Bressan,[a] Marianna Lucafò,[b] Vincenzo Petix,[b] Marina Busetti,[c] Lucilla Dolzani,[a] Cristina Lagatolla[a]

ᵃDepartment of Life Sciences, University of Trieste, Trieste, Italy
ᵇInstitute for Maternal and Child Health—IRCCS Burlo Garofolo, Trieste, Italy
ᶜMicrobiology Unit, University Hospital of Trieste, Trieste, Italy

**ABSTRACT** Carbapenem resistance in *Pseudomonas aeruginosa* strains responsible for chronic lung infections in cystic fibrosis (CF) patients is mainly due to loss of the OprD protein and, limited to meropenem and doripenem, to overexpression of efflux pumps. However, recent reports of isolates showing inconsistent genotype-phenotype combinations (e.g., susceptibility in the presence of resistance determinants and vice versa) suggest the involvement of additional factors whose role is not yet fully elucidated. Among them, the OpdP porin as an alternative route of entry for carbapenems other than OprD and the overexpression of two chromosomal carbapenemases, the *Pseudomonas*-derived cephalosporinase (PDC) and the PoxB oxacillinase, have recently been reconsidered and studied in specific model strains. Here, the contribution of these factors was investigated by comparing different phenotypic variants of three strains collected from the sputum of colonized CF patients. Carbapenem uptake through OpdP was investigated both at the functional level, by assessing the competition exerted by glycine-glutamate, the OpdP's natural substrate, against imipenem uptake, and at the molecular level, by comparing the expression levels of *opdP* genes by quantitative real-time PCR (qRT-PCR). Moreover, overexpression of the chromosomal carbapenemases in some of the isolates was also investigated by qRT-PCR. The results showed that, even if OprD inactivation remains the most important determinant of carbapenem resistance in strains infecting the CF lung, the interplay of other determinants might have a nonnegligible impact on bacterial susceptibility, being able to modify the phenotype of part of the population and consequently complicating the choice of an appropriate therapy.

**IMPORTANCE** This study examines the interplay of multiple factors in determining a pattern of resistance or susceptibility to carbapenems in clinical isolates of *Pseudomonas aeruginosa*, focusing on the role of previously poorly understood determinants. In particular, the impact of carbapenem permeability through OprD and OpdP porins was analyzed, as well as the activity of the chromosomal carbapenemases AmpC and PoxB, going beyond the simple identification of resistance determinants encoded by each isolate. Indeed, analysis of the expression levels of these determinants provides a new approach to determine the contribution of each factor, both individually and in coexistence with the other factors. The study contributes to understanding some phenotype-genotype discordances closely related to the heteroresistance frequently detected in *P. aeruginosa* isolates responsible for pulmonary infections in cystic fibrosis patients, which complicates the choice of an appropriate patient-specific therapy.

**KEYWORDS** OpdP porin, OprD, PDC AmpC variants, PoxB oxacillinase

Address correspondence to Cristina Lagatolla, clagatolla@units.it.

With the alarming spread of multidrug-resistant strains of *Pseudomonas aeruginosa*, identification of an adequate antimicrobial treatment has become a serious challenge. Special attention must be paid to patients affected by cystic fibrosis (CF), as their lungs are an ideal environment for colonization and long-term persistence of *P. aeruginosa*. This bacterium has been proven capable of rapidly adapting to these environmental conditions by employing a number of mechanisms, which include both the on-off switching of various genes and a high rate of genetic mutability. This leads to the evolution of the parental strain, with the emergence of different phenotypic variants that can exhibit high heterogeneity in terms of growth rate, colony morphology, ability to form biofilm, and antibiotic resistance (1–3). Carbapenems, along with tobramycin and colistin, remain the drugs of choice against *P. aeruginosa* respiratory infections, although this pathogen has demonstrated the ability to develop various resistance mechanisms. Modifications leading to ineffectiveness of carbapenems in *P. aeruginosa* chronically established in the CF lung consist mainly in the loss (or the expression of an inactive form) of the OprD outer-membrane porin, which is responsible for the uptake of basic amino acids (notably arginine) and carbapenem drugs into the bacterial cell, followed by the overexpression of efflux pumps, mainly MexAB-OprM (limited to meropenem and doripenem). On the other hand, carbapenemase production seems to be less frequent in CF *P. aeruginosa* isolates than in those collected from other clinical conditions (4–7). Besides these well-known and extensively reviewed resistance mechanisms (8, 9), some isolates with phenotype-genotype discordance have occasionally been described (10, 11), suggesting the involvement of additional, still cryptic factors whose role is not yet fully elucidated. For example, the loss of the OprD porin associated with a retained susceptibility to carbapenems has recently drawn attention to an alternative porin, OpdP, that is responsible for the uptake of glycine-glutamate (Gly-Glu) and involved in the compensatory uptake of arginine in OprD-defective strains (12). OpdP, also known as OccD3, exhibits 51% homology with OprD and has recently been considered for a possible role in carbapenem uptake (13, 14). On the other hand, carbapenem-resistant isolates carrying none of the above-mentioned resistance determinants have sometimes been documented. A recent study of the resistome of a *P. aeruginosa* population analyzed by whole-genome sequencing showed that about 9% of meropenem-resistant isolates fell into this category (10). In these strains, other, less common, genetic determinants that may be involved in the definition of the phenotype have been identified. These include the overproduction of variants of the AmpC cephalosporinase, named PDC (*Pseudomonas*-derived cephalosporinase), which carry specific mutations near the active site (e.g., the T105A substitution [with A replacing T at position 105]) that confer catalytic activity that extends to carbapenems (15). Also worth mentioning is the oxacillinase PoxB (OXA-50), first described by Kong et al. as a chromosomally encoded, noninducible carbapenemase, usually expressed at low levels and hence considered incapable of conferring carbapenem resistance (16). However, a recent study by Zincke et al. suggested that in the case of overexpression, this enzyme may reduce susceptibility to carbapenems, even if its relevance in a clinical context remains uncertain (17).

In the present study, we analyzed the expression of the resistance determinants described above in *P. aeruginosa* isolates collected from the sputum of CF patients. Different phenotypic variants of the same parental strain, showing different levels of resistance toward imipenem and meropenem, were compared, to better understand the interplay of the different mechanisms and their contributions to the final resistance phenotype of the clinical isolates.

## RESULTS

**Characterization of clinical isolates.** A small number of isolates of *P. aeruginosa* were collected from each of three chronically colonized CF patients. Typing by macrorestriction analysis showed that the pulsed-field gel electrophoresis (PFGE) profiles of

**TABLE 1** Phenotypic features of the different isolates of *P. aeruginosa*

| Strain | Isolate | PFGE type[a] | Colony morphology[b] | Biofilm production[c] | Carbapenem resistance pattern[d] | |
|---|---|---|---|---|---|---|
| | | | | | IPM | MEM |
| A | A1 | a | s/sl/p | + | S | S |
| | A2 | a | s/fa/p | N | R | I |
| | A3 | a | m/fa/np | ++ | R | I |
| B | B1 | b | m/fa/np | +++ | R | R |
| | B2 | b | s/fa/np | +++ | S | S |
| | B3 | b | s/fa/np | ++ | S | S |
| | B4 | b | r/sl/np | + | R | R |
| C | C1 | c | m/fa/np | + | S | I |
| | C2 | c | s/vsl/np | ++ | R | I |

[a]Different PFGE types are indicated by lowercase letters.
[b]Mucoidicity (s, smooth; m, mucoid; r, rough)/growth rate (fa, fast; sl, slow; vsl, very slow)/pigment production (p, pigment production; np, no pigment production).
[c]N, nonproducer; +, weak producer; ++, moderate producer; +++, strong producer (25).
[d]IPM, imipenem; MEM, meropenem; S, susceptible; I, intermediate; R, resistant.

isolates collected from the same patient were identical; in contrast, the profiles of isolates collected from different patients showed less than 68% similarity. Therefore, the three patients were assumed to be infected by unrelated strains, which were named A, B, and C, while isolates collected from the same patient, although showing different colony morphologies, were considered to belong to the same strain. Two to 4 isolates from each patient were selected, based on carbapenem resistance patterns, phenotypic differences in colony morphology, and biofilm production (Table 1), and analyzed for determinants contributing to their final phenotypes toward carbapenems.

First, different isolates of the same strain were compared for their ability to internalize the drugs, not only through the OprD porin but also through OpdP, already hypothesized as an alternate entry pathway for carbapenems (13, 14). In addition, overexpression of the MexAB-OprM efflux pump, which provides the extrusion of meropenem but not imipenem, was investigated. However, since this feature was equally expressed in isolates of the same strain (Table 2), this analysis was considered not useful for the aim of this study and imipenem was considered a better marker.

Later, in order to look for the production of inactivating enzymes, the isolates were analyzed using the modified carbapenem inactivation method. This assay excluded the production of the most common carbapenemases for all isolates, although in some cases, a zone diameter of 18 mm, which was just below the threshold of the assay and categorized as an indeterminate result, was obtained. Hence, we decided to consider the possible involvement of two chromosomally encoded enzymes, the PoxB oxacillinase and the PDC allelic variants of the AmpC beta-lactamase, for which low hydrolytic activity toward carbapenems has been described (15, 17). Indeed, sequencing of *ampC* genes followed by *in silico* translation revealed that, even if different variants of the AmpC enzyme were detected, all isolates carried the T105A substitution (Table 3), which is the most common in the PDC allelic variants and is strictly related to a reduced susceptibility to imipenem when paired with overexpression of the enzyme (15).

Below, a comparison of the different isolates is reported for each strain, in order to elucidate the contributions of the different determinants to the final phenotype.

**(i) Strain A.** *In silico* analysis of the *oprD* genes revealed that none of the three isolates selected for strain A expressed the porin, due to a 46-bp insertion (isolate A1) or to a single-base deletion (isolates A2 and A3) that resulted in the production of aberrant, nonfunctional peptides (Table 2). In the case of isolate A1, this finding looked inconsistent with its phenotype, which was susceptible, suggesting the existence of an alternative entry pathway for carbapenems, as already hypothesized (18). In order to

Microbiology Spectrum

**TABLE 2** Analysis of outer membrane protein involvement in carbapenem uptake or extrusion

| Strain | Isolate | Structure analysis of OprD | | Effect on OprD | Functional analysis (MIC [mg/liter]) of: | | | | MexAB-OprM |
| | | Gene mutation | | | OpdP: | | | | |
| | | Type | Sequence change | | IPM | IPM + Gly-Glu[a] | MEM | MEM + Gly-Glu[a] | MEM + PAβN[b] |
| A | A1 | Frame shift | $_{nt270}$46nt$_{nt271}$[c] | Porin loss | 1 | 4 | 2 | 2 | 0.5 |
| | A2 | Frame shift | $\Delta_{nt541-551}$ | Porin loss | 16 | 16 | 4 | 4 | 0.5 |
| | A3 | Frame shift | $\Delta_{nt541-551}$ | Porin loss | 8 | 64 | 4 | 4 | 0.5 |
| B | B1 | Frame shift | $_{nt709}$T$_{nt710}$ | Porin loss | 16 | 128 | 16 | 16 | 16 |
| | B2 | None | | Full length | 4 | 16 | 1 | 2 | 1 |
| | B3 | None | | Full length | 4 | 4 | 1 | 1 | 1 |
| | B4 | None | | Full length | 16 | 32 | 8 | 16 | 8 |
| C | C1 | Frame shift | $_{nt428}$GA$_{nt429}$ | Porin loss | 2 | 16 | 4 | 8 | 1 |
| | C2 | STOP | $_{nt414}$TGG→TGA | L1-L2 only | 16 | 32 | 4 | 4 | 1 |

[a]Carbapenem uptake through OpdP: gray shading indicates 4- or 8-fold MIC increase.
[b]Meropenem extrusion by MexAB-OprM: gray shading indicates 4-fold MIC decrease.
[c]46nt, 46-nucleotide insertion AGCTCGACGGCACCTCCGACAAGACCGGCACCGGCAACCTGCCGGT.

investigate whether the OpdP porin was involved in this uptake, the carbapenem susceptibilities of the different isolates were evaluated after the addition of 10 mM Gly-Glu, the OpdP's natural substrate. The addition of the competitor led to different results in the three isolates, with a strong effect especially on imipenem resistance that reached 4- and 8-fold increases in A1 and A3, respectively. In contrast, the phenotype of A2 was unaffected (Table 2), suggesting that this isolate either produced a mutated, less functional OpdP or expressed the porin to a lesser extent. Sequencing of the *opdP* gene revealed no difference: the same allelic form, showing a perfect homology with many strains already reported in public databases (e.g., see the nucleotide sequence with GenBank accession number LR590473), was detected not only in the three isolates of strain A but also in all isolates from this study. In contrast, different expression levels were observed: no overexpression was detected in isolate A2 compared to the expression level in the reference strain PAO1, while significant *opdP* overexpression was detected in isolates A1 and A3 (Fig. 1). This result could explain the imipenem susceptibility of isolate A1 but left unsolved the question of A3, which looked resistant despite an even higher level of *opdP* overexpression than in A1. Hence, the activity of the two chromosomally encoded carbapenemases was considered by evaluating the expression levels of *poxB* and *ampC*. In isolates A1 and A2, quantitative real-time PCR (qRT-PCR) analysis showed that the expression of these enzymes was comparable to that of PAO1. In contrast, significant overexpression of both (approximately 15-fold higher than in PAO1) was detected in isolate A3, indicating that, in this case, hydrolysis of the drug predominated over its uptake and contributed significantly to the expression of a resistant phenotype.

**TABLE 3** PDC allelic variants of AmpC beta-lactamase identified in the nine isolates of *P. aeruginosa*[a]

| Strain | Isolate | Amino acid substitutions | Allelic form |
| --- | --- | --- | --- |
| A | A1 | T105A V205L | PDC[b] |
| | A2 | T105A V205L | PDC[b] |
| | A3 | T105A V205L | PDC[b] |
| B | B1 | R79Q T105A | PDC-5 |
| | B2 | R79Q T105A | PDC-5 |
| | B3 | T105A L176R | PDC-8 |
| | B4 | P23L R79Q T105A V239A | PDC[b] |
| C | C1 | T105A L176R | PDC-8 |
| | C2 | T105A L176R | PDC-8 |

[a]PDC, *Pseudomonas*-derived cephalosporinase (15).
[b]Not previously described.

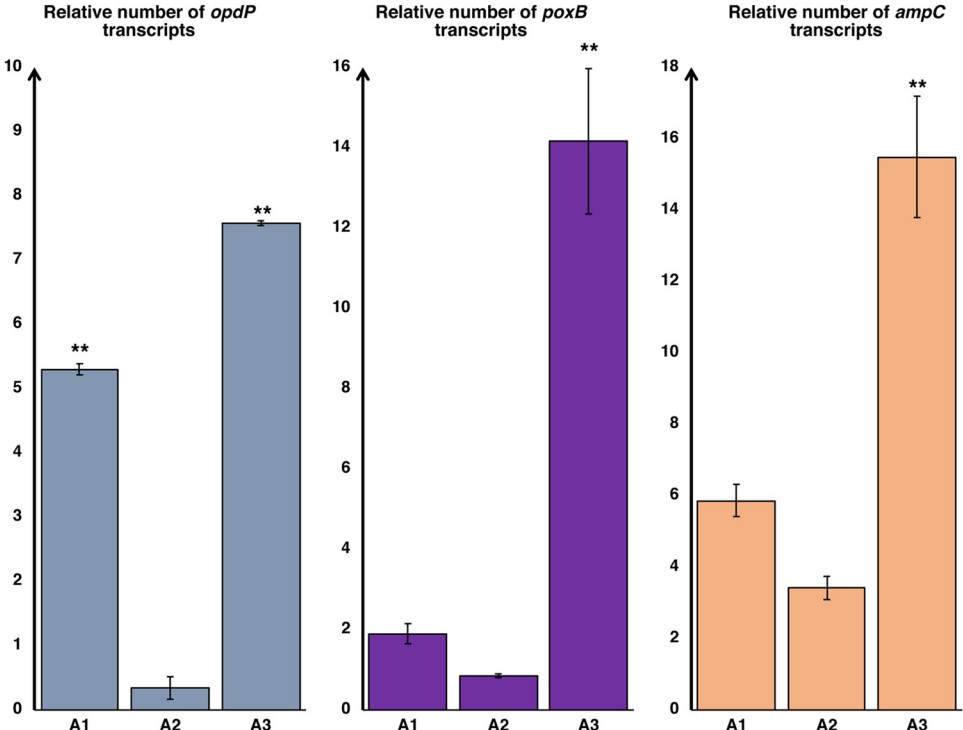

**FIG 1** Relative expression levels of *opdP*, *ampC*, and *poxB* (with respect to control strain PAO1) in strain A. Values were normalized using the expression of housekeeping gene *rpoD*. Error bars show standard deviations. **, $P < 0.05$ (as determined by one-way analysis of variance [ANOVA] and Bonferroni *post hoc* test).

**(ii) Strain B.** Of the four isolates from strain B, only isolate B1 carried an inactivated *oprD* gene, due to a single-base insertion that resulted in the production of an aberrant protein (Table 2). Further analysis of this isolate revealed an 8-fold increase in the imipenem MIC after the addition of Gly-Glu (Table 2), according with the significant overexpression of *opdP* (Fig. 2) and indicating an even higher uptake of the drug through the OpdP porin than in the isolates described above. However, the production of significant amounts of both PoxB and AmpC carbapenemases was also detected in this isolate (Fig. 2), suggesting that its resistant phenotype is due to the prevalence of drug hydrolysis over its uptake, albeit abundant, via OpdP.

The remaining three isolates from strain B carried an intact *oprD* gene that, compared to its homolog in PAO1, encoded a porin that exhibited a divergent stretch of 10 amino acid residues in the L7 loop. Notably, this allelic form was associated with an increased meropenem susceptibility (which was indeed detected in these isolates compared to their imipenem susceptibility), due to a modification of the porin channel that facilitates entry of this drug (19). To obtain a complete picture of carbapenem uptake, the expression levels of *oprD* were also evaluated in these isolates and analyzed together with the expression levels of *opdP*. Actually, isolate B4 showed low expression levels of both porins, which combined with significant overexpression of both carbapenemases, explained its resistant phenotype. Conversely, the susceptibility of isolates B2 and B3 was consistent with their effective uptake of the drug, especially through OprD, associated with low production of carbapenemases (Fig. 2).

**(iii) Strain C.** Isolate C1 was found to be similar to isolate A1, with a susceptible phenotype that was inconsistent with the loss of the OprD porin, which was detected by *oprD* sequencing. Subsequent studies of this isolate strongly supported the hypothesis of a major contribution of the OpdP porin in determining the final phenotype. Indeed, the imipenem MIC increased 8-fold after the addition of 10 mmol Gly-Glu, and qRT-PCR showed the highest overexpression of the *opdP* gene among the isolates considered in

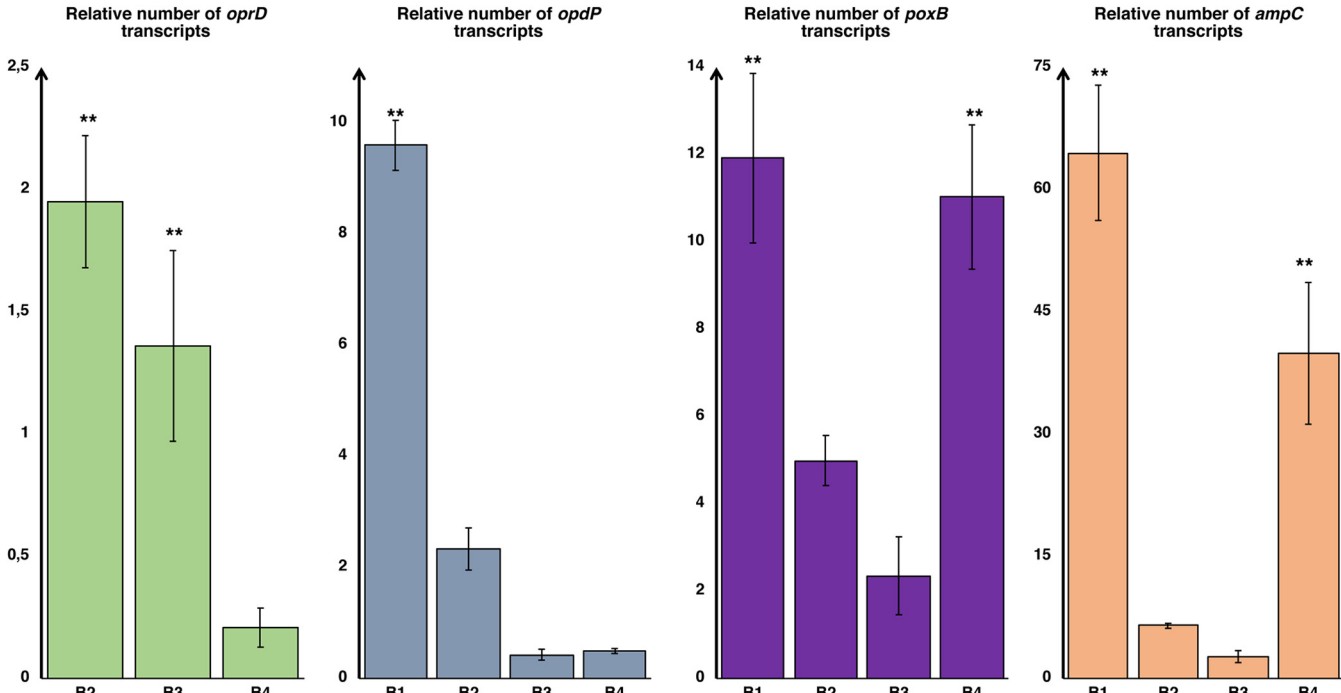

**FIG 2** Relative expression levels of *oprD*, *opdP*, *ampC*, and *poxB* (with respect to control strain PAO1) in strain B. Values were normalized using the expression of housekeeping gene *rpoD*. Error bars show standard deviations. **, $P < 0.05$ (as determined by one-way ANOVA and Bonferroni *post hoc* test).

this study (Table 2 and Fig. 3). Thus, the susceptible phenotype exhibited by isolate C1, despite the simultaneous overexpression of both chromosomal carbapenemases (Fig. 3), suggests that in this isolate, the entry of carbapenems through the OpdP channel reaches very substantial levels.

In contrast, in isolate C2, the production of a truncated OprD due to a point mutation that introduced a premature stop codon into the *oprD* gene (Table 2) was not counterbalanced by *opdP* overexpression (Fig. 3). Hence, also considering that the two chromosomal carbapenemases were not overexpressed in this isolate, the lack of carbapenem uptake was considered the major cause of its resistant phenotype.

## DISCUSSION

Carbapenem resistance in *P. aeruginosa* strains that are responsible for chronic pulmonary infections in cystic fibrosis patients is frequently associated with inactivation of the OprD outer membrane porin and, limited to meropenem and doripenem, with overexpression of efflux pumps. However, occasional reports of strains expressing phenotypes that are inconsistent with their genotypes (10, 11) suggest that additional, still cryptic, mechanisms may be involved. In particular, the possible role of the OpdP porin as an alternative route of entry for carbapenems (12–14) and the overproduction of two chromosomally encoded carbapenemases (15, 20) have recently been reconsidered in studies of specific mutant strains generated by site-directed mutagenesis (14, 17). Here, the phenotypes of different variants of three clinical isolates were studied in relation to the expression levels of their resistance determinants, in order to evaluate the actual contributions of these factors. With this aim, we first considered susceptibility to both imipenem and meropenem. However, after finding no difference in the expression levels of the MexAB-OprM efflux pump between isolates of the same strain, we realized that, at least in two of three isolates from this study, the overexpression of this efflux system complicated the interpretation of meropenem resistance, so imipenem was considered a better marker.

Analysis of isolates A1 and C1 supported the hypothesis that an effective carbapenem uptake through OpdP was the most likely explanation for their susceptible

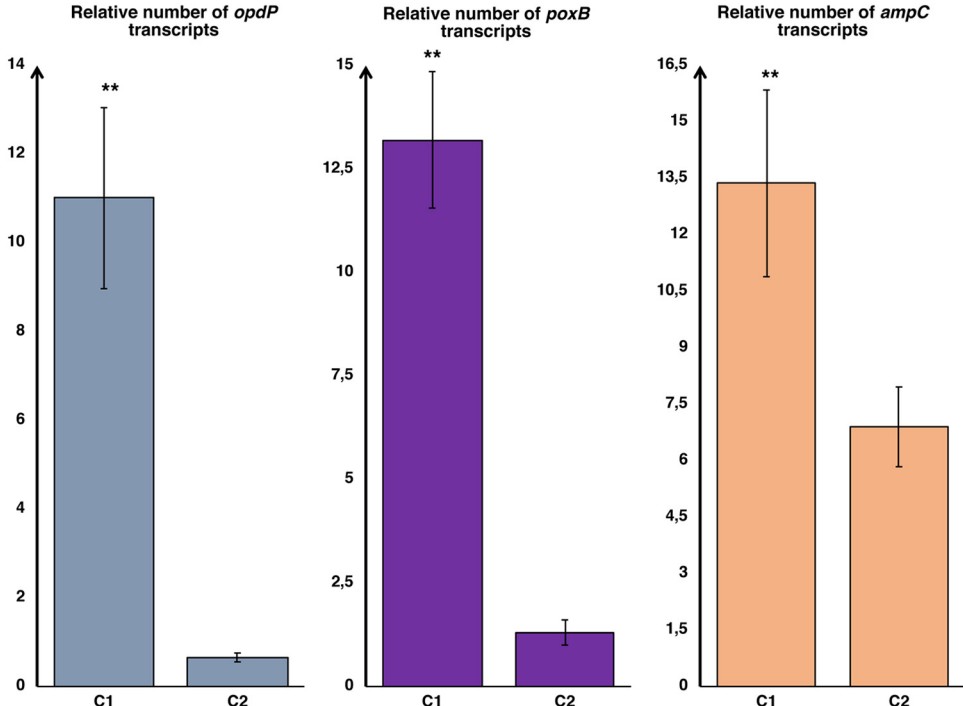

**FIG 3** Relative expression levels of *opdP*, *ampC*, and *poxB* (with respect to control strain PAO1) in strain C. Values were normalized using the expression of housekeeping gene *rpoD*. Error bars show standard deviations. \*\*, $P < 0.01$ (as determined by Student's *t* test).

phenotype. In the case of strain A, isolates A1 and A2 were both characterized by loss of the OprD porin and by the production of negligible amounts of chromosomal carbapenemases (Fig. 1). They differed only in the decrease of imipenem uptake in the presence of Gly-Glu, which was observed in A1, which indeed showed a susceptible phenotype, and was not detected in A2, which was resistant (Table 2). The comparison between the two isolates of strain C led to the same conclusion, to even a greater extent since the high *opdP* expression level detected in C1 (the highest of this study) conferred a susceptible phenotype despite the overexpression of PoxB and AmpC carbapenemases. Molecular analysis provided interesting results on the minimal amount of *opdP* gene expression required to detect porin functionality. In four isolates that did not significantly modify their phenotypes after the addition of Gly-Glu, *opdP* was underexpressed compared to its expression in the control strain. In contrast, in the remaining isolates, overexpression of *opdP* was linked to a significant increase in imipenem resistance after the addition of Gly-Glu. In conclusion, these data support the hypothesis that OpdP, even when OprD is not expressed, can provide the uptake of a sufficient amount of imipenem to make the strain susceptible, although it remains difficult to establish the minimum level of expression required to confer susceptibility, as it is strictly dependent on the concurrent expression of other resistance mechanisms. We are aware that the definitive proof of the effective contribution of the OpdP porin to the final phenotype would come from inactivating the *opdP* gene in the A1 or C1 isolate. Unfortunately, this was not possible, because the multidrug resistance of these isolates made it impossible to perform knockout experiments.

On the other hand, an effective contribution was also inferred for the chromosomal carbapenemases. Regarding isolates A1 and A3, the main difference was actually the overexpression of these enzymes in the latter, indicating that their contribution to the establishment of a resistant phenotype is not negligible, although the hydrolytic activity of these enzymes toward carbapenems is generally considered to be low. Similarly,

the comparison between isolates B3 and B4 led to the same conclusion, although in this case, in addition to the different expression levels of the two carbapenemases, B4 was also found to underexpress the *oprD* gene, which made the interpretation somewhat more difficult. In fact, the imipenem resistance of B4 must be considered a result of the abundant production of both the PoxB oxacillinase and the PDC variant of AmpC, which are about 5 and 20 times higher, respectively, than in B3, combined with poor drug uptake due to the production of a functional but low-abundance OprD and to the concurrent absence of OpdP.

For other pairs of isolates, the direct comparison was even more complicated, since in the presence of coexpression of different determinants, the precise contribution of each of them is rather difficult to assess, as it depends not only on their mere absence or presence but, more importantly, on their expression levels. Nevertheless, our data support the hypothesis that, even if *oprD* inactivation remains by far the most important determinant conferring imipenem resistance in *P. aeruginosa* strains infecting the CF lung, mutations leading to differential expression of the OpdP porin and chromosomal carbapenemases actually occur during the course of chronic infections, modifying the phenotype of a portion of the bacterial population and contributing to the generation of heteroresistance that is often responsible for the lack of concordance between susceptibility testing and clinical outcome (8).

Evidence that these determinants are indeed effective in the clinical setting should be taken into account in the development of new therapeutic compounds. The design of carbapenems that act as a preferential substrate for the OpdP porin or of specific inhibitors against the two chromosomal carbapenemases could provide further therapeutic options against the different phenotypic variants of *P. aeruginosa* that colonize the CF lung and allow better management of chronic infections in these patients.

## MATERIALS AND METHODS

**Bacterial strains.** *P. aeruginosa* isolates were originally collected for diagnostic purposes from the sputum of three chronically infected CF patients in care at the Regional Center for Cystic Fibrosis in Trieste and provided to investigators within a protocol approved by the local institutions' institutional Review Board. Plates were incubated for 72 to 96 h at 37°C and visually inspected. Various numbers of colonies (2 to 10) were reisolated from each sputum sample, trying to pick representatives of all different morphotypes. Bacterial identification was performed using a Vitek-2 system (bioMérieux, Marcy-l'Etoile, France).

Clonal relatedness between the different isolates was investigated by macrorestriction analysis as previously described (21). Briefly, genomic DNA was digested with SpeI (Merck KGaA, Darmstadt, Germany). Pulsed-field gel electrophoresis (PFGE) of DNA fragments was performed in a CHEF DR III apparatus (Bio-Rad) at 14°C and 6 V/cm for 26 h using pulse times ranging from 1 to 25 s and a 120° switch angle. DNA patterns were analyzed with the GelCompar II version 6.6 software (Applied-Maths, Kortrijk, Belgium), using the Dice coefficient for pairwise comparison. Strains with pattern similarities of <90% were considered not related. Different PFGE types are indicated by lowercase letters (Table 1).

**Susceptibility testing.** Antimicrobial susceptibility was evaluated routinely by the disk diffusion method (22) and interpreted according to CLSI guidelines. *P. aeruginosa* ATCC 27853 was used as a control strain.

MICs of imipenem and meropenem were determined by microbroth dilutions in cation-adjusted Mueller-Hinton broth according to Clinical and Laboratory Standards Institute guidelines (22). A concentration of 10 mmol glycine-glutamate (Gly-Glu), the natural OpdP porin substrate, was added to evaluate carbapenem uptake through this alternative porin (23). Besides this, the meropenem MICs were evaluated after the addition of 20 mg/liter of the efflux pump inhibitor Phe-Arg $\beta$-napthylamide (PA$\beta$N) (23), to assess the contribution of the MexAB-OprM efflux pump to meropenem resistance.

Carbapenemase production was investigated by the modified carbapenem inactivation method as recommended by CLSI. Briefly, a 10-$\mu$g meropenem disk was added to 2 ml of Trypticase soy broth inoculated with the test bacterium, incubated for 4 h at 35°C, and then placed on a Mueller-Hinton agar plate inoculated with *Escherichia coli* ATCC 25922 according to the routine disk diffusion procedure. The results were interpreted as follows: carbapenemase positive with a zone diameter of ≤15 mm, carbapenemase negative with a zone diameter of ≥19 mm, and an indeterminate result represented by an intermediate zone (16 to 18 mm) (22).

**Evaluation of BF production.** The amount of biofilm (BF) produced was evaluated in 96-well microtiter plates (Nunc; Thermo Fisher Scientific, Paisley, UK) by crystal violet staining as previously described (24). Briefly, 200 $\mu$l of an overnight culture, previously diluted 1,000× in cation-adjusted Mueller-Hinton

**TABLE 4** Primers used in this work

| Primer | Nucleotide sequence | Use | Reference or source |
|---|---|---|---|
| D-Fw | 5'-ATGCGACATGCGTCATGCAAT-3' | Amplification and sequencing of *oprD* genes | 20 |
| D-Rev | 5'-CGGTACCTACGCCCTTCCTT-3' | | |
| opdP-Fw | 5'-GAGCAATCAGGTGATGAGAA-3' | Amplification and sequencing of *opdP* genes | This study |
| opdP-Rev | 5'-GCAGGTTTACAGCAGGTT-3' | | |
| opdP-mid-Fw | 5'-CTGCCCTCCAGCTTCAC-3' | Sequencing of internal fragment of *opdP* genes | This study |
| opdP-mid-Rev | 5'-GGTCCGCGGGCTGAC-3' | | |
| PreAmpC-PA1 | 5'-ATGCAGCCAACGACAAAGG-3' | Amplification and sequencing of *ampC* genes | 15 |
| PostAmpC-PA2 | 5'-CGCCCTCGCGAGCGCGCTTC-3' | | |
| qRT-oprD-Fw | 5'-AAGTGATGAAGTGGAGCG-3' | Quantitative real-time PCR of *oprD* genes | This study |
| qRT-oprD-Rev | 5'-TCGCTTCGGCCTGA-3' | | |
| qRT-opdP-Fw | 5'-ACAGCTTCACCTTCCGCAT-3' | Quantitative real-time PCR of *opdP* genes | This study |
| qRT-opdP-Rev | 5'-AGCCCGAGCTGTACTTGAG-3' | | |
| qRT-ampC-Fw | 5'-CGCCGTACAACCGGTGAT-3' | Quantitative real-time PCR of *ampC* genes | 26 |
| qRT-ampC-Rev | 5'-CGGCCGTCCTCTTTCGA-3' | | |
| qRT-poxB-Fw | 5'-AATCGGCCAGGTTGTGGATAA-3' | Quantitative real-time PCR of *poxB* genes | 17 |
| qRT-poxB-Rev | 5'-GGAGCAGAAAGCGGGTCTGT-3' | | |
| qRT-rpoD-Fw | 5'-GGGCTGTCTCGAATACGTTGA-3' | Quantitative real-time PCR of *rpoD* genes | 26 |
| qRT-rpoD-Rev | 5'-ACCTGCCGGAGGATATTTCC-3' | | |

broth, was inoculated into each well (each sample in triplicate). After 24 h of static incubation at 37°C, the broth culture was removed, each well was gently rinsed with sterile saline, and BF was fixed by incubating the plate at 60°C for 1 h. Fixed BF was stained for 15 min with 250 $\mu$l of 2% crystal violet, and the plates were rinsed with water and air dried. The amount of crystal violet bound to BF was evaluated by measuring the optical density (OD) at 570 nm after solubilization in 300 $\mu$l of 33% acetic acid for 30 min. A cutoff value ($OD_{cut}$) was established for each experiment, defined as three standard deviations above the mean OD ($OD_{avg}$) of the negative control, i.e., $OD_{cut} = OD_{avg}$ of negative control + (3 × standard deviation of OD of negative control).

Results were interpreted following the criteria described by Stepanović et al. (25): nonproducer (N), $OD \leq OD_{cut}$; weak producer (+), $OD_{cut} < OD \leq 2 \times OD_{cut}$; moderate producer (++), $2 \times OD_{cut} < OD \leq 4 \times OD_{cut}$; and strong producer (+++), $OD > 4 \times OD_{cut}$.

**PCR amplification and DNA sequencing.** Sequencing of *oprD*, *opdP*, and *ampC* genes of the different isolates was performed to identify mutations that significantly affected their functionality. The three genes were amplified using primers listed in Table 4 and conditions previously described (see references in Table 4). Sequencing reactions were carried out at a commercial sequence facility (BMR Genomics, Padua, Italy). The nucleotide and protein sequences were analyzed using the *blastn*, *blastp*, and *bl2seq* algorithms available at the National Center of Biotechnology Information website (http://www.ncbi.nlm.nih.gov).

**Gene expression analysis.** Quantitative real-time PCR (qRT-PCR) was used to measure the expression of the *opdP*, *oprD*, *ampC*, and *poxB* genes, using specific oligonucleotide primers listed in Table 4. Total RNA of each isolate was extracted using the PureLink RNA minikit (Thermo Scientific, Waltham, MA, USA) according to the manufacturer's recommendations, resuspended in 30 to 40 $\mu$l of RNase-/DNase-free water, and quantified by using a NanoDrop 2000 spectrophotometer (Thermo Scientific). The equivalent of 1 $\mu$g RNA was reverse transcribed using the high-capacity RNA-to-cDNA kit (Applied Biosystems, Foster City, CA, USA). The reaction was carried out at 37°C for 60 min and stopped by heating to 95°C for 5 min. The cDNA obtained was diluted in RNase-/DNase-free water and amplified by quantitative PCR using the KICqStart SYBR green qPCR ready mix (Sigma-Aldrich, Saint Louis, MO, USA). The reaction mixture for each sample consisted of 1× reaction buffer, 1 $\mu$g/ml cDNA, and 20 $\mu$M gene-specific primers in a final volume of 20 $\mu$l. The qRT-PCR was performed in a C1000 thermal cycler (Applied Biosystems) with the following amplification conditions: a single cycle at 95°C for 5 min and 40 cycles at 95°C for 30 s, 64°C for 30 s, and 72°C for 3 s. A control reaction was performed for each sample by using the original RNA in the reaction mixture to verify the absence of residual DNA. Experiments were reproduced at least in triplicate for each target gene. Normalization was performed with the *rpoD* gene as an internal standard (26), using the comparative cycle threshold ($C_T$) method (27), and the values obtained were then compared to the ones obtained with the reference strain PAO1 ($2^{-\Delta\Delta CT}$ method). Values greater than 1 were labeled as "overexpression," while values lower than 1 were labeled as "underexpression."

## ACKNOWLEDGMENT

This research did not receive any specific grant from funding agencies in the public, commercial, or not-for-profit sectors.

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
