## [Reviewer comments · Microbiology Spectrum]

Microbiology Spectrum

Interplay of OpdP porin and chromosomal carbapenemases in the determination of carbapenem resistance/susceptibility in *Pseudomonas aeruginosa*.

Jad Atrissi, Annalisa Milan, Raffaella Bressan, Marianna Lucafò, Vincenzo Petix, Marina Busetti, Lucilla Dolzani, and Cristina Lagatolla

Corresponding Author(s): Cristina Lagatolla, University of Trieste

Review Timeline:

Submission Date:

August 12, 2021

Accepted:

August 23, 2021

Editor: Ayush Kumar

Reviewer(s): The reviewers have opted to remain anonymous.

Transaction Report:

DOI: <https://doi.org/10.1128/Spectrum.01186-21>

August 23, 2021

Dr. Cristina Lagatolla
University of Trieste
Life Sciences
Via Fleming 22
Trieste 34127
Italy

Re: Spectrum01186-21 (**Interplay of OpdP porin and chromosomal carbapenemases in the determination of carbapenem resistance/susceptibility in *Pseudomonas aeruginosa*.**)

Dear Dr. Cristina Lagatolla:

I have reviewed the revised version of the manuscript that was initially submitted to AAC. I am satisfied by the changes you have made to the manuscript in response to the reviewers' comments from your previous submission. Thus, your manuscript has been accepted, and I am forwarding it to the ASM Journals Department for publication. You will be notified when your proofs are ready to be viewed.

Sincerely,

Ayush Kumar
Editor, Microbiology Spectrum
